# High-fidelity endonuclease variant HypaCas9 facilitates accurate allele-specific gene modification in mouse zygotes

Arisa Ikeda[1], Wataru Fujii 🄳 [1]*, Koji Sugiura 🄳 [1] & Kunihiko Naito[1]

CRISPR/Cas9 has been widely used for the efficient generation of genetically modified animals; however, this system could have unexpected off-target effects. In the present study, we confirmed the validity of a high-fidelity Cas9 variant, HypaCas9, for accurate genome editing in mouse zygotes. HypaCas9 efficiently modified the target locus while minimizing off-target effects even in a single-nucleotide mismatched sequence. Furthermore, by applying Hypa-Cas9 to the discrimination of SNP in hybrid strain-derived zygotes, we accomplished allele-specific gene modifications and successfully generated mice with a monoallelic mutation in an essential gene. These results suggest that the improved accuracy of HypaCas9 facilitates the generation of genetically modified animals.

[1] Department of Animal Resource Sciences, Graduate School of Agricultural and Life Sciences, The University of Tokyo, 1-1-1 Yayoi, Bunkyo-ku, Tokyo 113-8657, Japan. *email: awtrfj@mail.ecc.u-tokyo.ac.jp

CRISPR/Cas9-mediated gene editing in zygotes enables systemic genome modification in various animal species[1–10]. CRISPR/Cas9 consists of Cas9 endonuclease and single-guide RNA (gRNA). The Cas9–gRNA complex binds to the target DNA sequence complementary to the protospacer of gRNA and introduces double-strand breaks in DNA, and eventually insertions and deletions (indels) are induced into the target locus via error-prone nonhomologous end-joining[11–13]. However, the Cas9–gRNA complex sometimes tolerates several nucleotide mismatches in the gRNA–DNA heteroduplex and induces unwanted mutations at off-target sites[5,11,12]. It has been reported that the use of paired Cas9 nickase or FokI-dCas9 can reduce the off-target effects, but they have the limitations of designable loci because they require the design of gRNAs on the plus- and minus-strands within a limited distance[14–17]. Therefore, to retain the diversity of targetable loci comparable to wildtype (WT)-Cas9, it is necessary to improve the accuracy of an approach that requires only a single gRNA for each target locus. Cas9 variants whose accuracy of target recognition is enhanced by amino-acid substitutions have been screened[18–24]. One of these variants, hyper-accurate (Hypa) Cas9, exhibited significantly higher accuracy than WT-Cas9 and minimized off-target cleavage[20]. However, such accuracy has been tested only in limited types of cultured cells, and it remains unknown whether Hypa-Cas9 works effectively in mammalian zygotes.

In the present study, we investigated the validity of HypaCas9 for accurate genome editing in mouse zygotes. The results showed that HypaCas9 could reduce off-target effects in mouse zygotes and the accuracy of HypaCas9 enables the discrimination of single-nucleotide polymorphisms (SNPs) and the introduction of monoallelic mutations, and thereby the efficient generation of genetically modified mice targeting the genes essential for survival

and development. Our finding suggests that the improved accuracy of HypaCas9 facilitates the generation of genetically modified animals.

## Results

**Evaluation of the accuracy of HypaCas9 in mouse zygotes.** HypaCas9 has enhanced specificity for target DNA because of the N692A, M694A, Q695A, and H698A substitutions in its REC3 domain (Supplementary Fig. 1A), which acts as an allosteric effector of its nuclease domain[20]. We constructed a HypaCas9 vector by introducing the alanine substitutions into the previously reported WT-Cas9 vector, which is optimized for expression in mouse zygotes by addition of the 3′UTR sequence of Tbpl1 and a 95nt polyadenine tail sequence[5]. It was confirmed that the substitutions did not affect the level of protein expression (Supplementary Figs. 1B and 11A).

Then, we evaluated the accuracy of HypaCas9 in mouse zygotes. To compare the specificity for target DNA between WT-Cas9 and HypaCas9, we utilized Gt(ROSA)26Sor-gRNA, which was previously reported to exhibit substantial off-target effects[5] (Fig. 1a), and assessed whether Cas9–gRNA introduced mutation into the on- and off-target locus by Sanger sequencing of polymerase chain reaction (PCR) products in each embryo at the blastocyst stage (Fig. 1b and Supplementary Fig. 2). Whereas WT-Cas9 introduced on- and off-target mutations in almost all the embryos analyzed (100% and 93.3%, respectively), HypaCas9 introduced no mutations in the off-target site and also retained its on-target mutation efficiency, suggesting that HypaCas9 recognizes the target DNA sequence more precisely and reduces the off-target risk in the mouse zygote. This off-target site has two nucleotide mismatches against gRNA, and thus we examined whether HypaCas9 could discriminate even a single mismatched

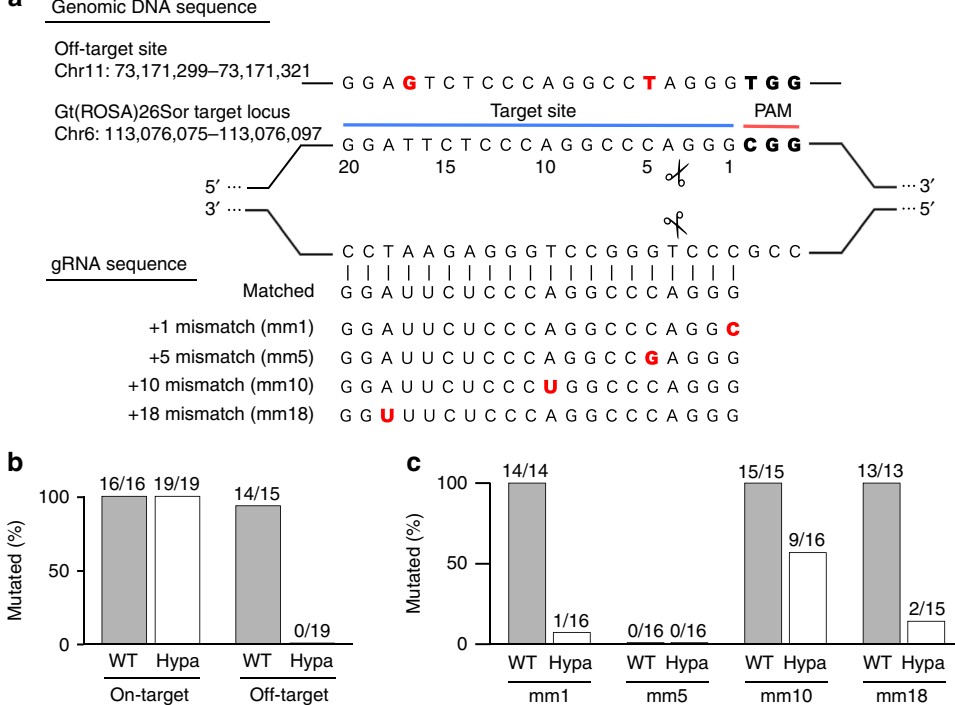

**Fig. 1** HypaCas9 exhibits reduced off-target effects without compromising on-target activity in the mouse zygote. **a** Genomic DNA sequence of the Gt(ROSA)26Sor-gRNA targeting site and off-target site, and the design of matched and singly mismatched gRNAs. The mismatched nucleotides are shown in red. The locus information is based on GRCm38.p4. **b** Comparison of on- and off-target activity between WT-Cas9 and HypaCas9 in the mouse zygote. Cas9 mRNA was injected into zygotes with Gt(ROSA)26Sor-targeting matched gRNA and the mutation efficiency was evaluated at the blastocyst stage. **c** Evaluation of single mismatch tolerance of WT-Cas9 and HypaCas9. gRNA with a single mismatch at position 1, 5, 10, or 18 was injected into mouse zygotes. In (**b** and **c**), numbers above the bar indicate the numbers of mutated embryos per total embryos from two replicate experiments

base pair, using gRNAs with single mismatches at positions 1, 5, 10, or 18 (Fig. 1a). WT-Cas9 mutated the target site in all the embryos when combined with +1-, 10-, or 18-mismatched gRNA. By contrast, HypaCas9 activity was reduced by a single mismatch at all the positions, although the mutation was detected in almost half of the embryos when using +10-mismatched gRNA (Fig. 1c and Supplementary Fig. 3). To validate the accuracy of HypaCas9 at another locus, we performed the evaluation at the *Tyrosinase* (*Tyr*) locus (Supplementary Fig. 4). *Tyr*-gRNA1 was introduced into mouse zygotes with WT-Cas9 or HypaCas9, and the mutation rate was analyzed at the on-target locus and two off-target loci (Supplementary Fig. 4A). As with *Gt(ROSA)26Sor*-gRNA, HypaCas9 showed a reduced off-target mutation rate at both off-target sites 1 and 2 with *Tyr*-gRNA1, while introducing mutation at the on-target site in all the embryos (Supplementary Figs. 4B and 5). In addition, we utilized matched and singly mismatched *Tyr*-gRNA2 (Supplementary Fig. 4C), and showed that off-target mutation by HypaCas9 was suppressed when using +1, 5, 10, 14, or 18-mismatched *Tyr*-gRNA2 (Supplementary Figs. 4D and 6). These results suggest that the improved specificity of HypaCas9 enables accurate targeting sufficient to discriminate a single mismatch in the mouse zygotes.

**Introduction of allele-specific mutations using HypaCas9.** Next, we attempted to apply the high accuracy of HypaCas9 to allele-specific gene editing by utilizing an SNP between the mouse strains. The endonuclease activity of WT-Cas9 is precisely suppressed when SNP or nucleotide substitution disrupts the protospacer adjacent motif (PAM)[25,26]. By contrast, when a mismatched nucleotide is positioned in the protospacer of gRNA, the off-target effect of WT-Cas9 can introduce mutation into an untargeted allele with a single mismatch, as indicated in previous studies[11,12] and by the results in Fig. 1. To test whether the accuracy of HypaCas9 facilitates SNP discrimination in this approach, we designed two gRNAs targeting the *Cdx2* exon 3 locus, which has an SNP (NCBI dbSNP Build 142; rs32379528: C57BL/6N (B6), G/G; DBA/2J (D2), C/C) (Fig. 2a). Only in the B6 allele, PAM for gRNA1 is present and gRNA2 shows perfect complementarity. Each of these B6-targeting gRNAs was injected with WT-Cas9 or HypaCas9 into B6D2F1 zygotes, and the alleles were separated from each other and sequenced independently (Fig. 2b). The allele separation was carried out by PCR restriction fragment length polymorphism using an SNP (NCBI dbSNP Build 142; rs29679715: B6, T/T; D2, A/A) (Supplementary Figs. 7 and 11B). HindIII cleaves only the D2 allele at the SNP locus, leading to the allele separation in electrophoresis. In this analysis, WT-Cas9 exhibited the introduced mutation specific for the B6 allele when coinjected with gRNA1, but exhibited mutations for both the B6 and D2 alleles when using gRNA2, indicating that WT-Cas9 failed to discriminate the single-nucleotide mismatch in the D2 allele (Table 1 and Supplementary Fig. 8). By contrast, HypaCas9 could introduce the B6 allele-specific mutations in all embryos, even with gRNA2, suggesting that HypaCas9 successfully discriminated the untargeted allele with a single mismatch in a highly specific manner in contrast to WT-Cas9, and the accuracy of HypaCas9 expands the applicable loci of allele-specific gene modification in the mouse zygotes.

**Generation of monoallelic gene-modified mice using HypaCas9.** Finally, we applied HypaCas9 to generate allele-specific genome-modified mice targeting the gene essential for survival. When targeting such genes, in which biallelic mutation results in lethality, the deterministic monoallelic gene modification using

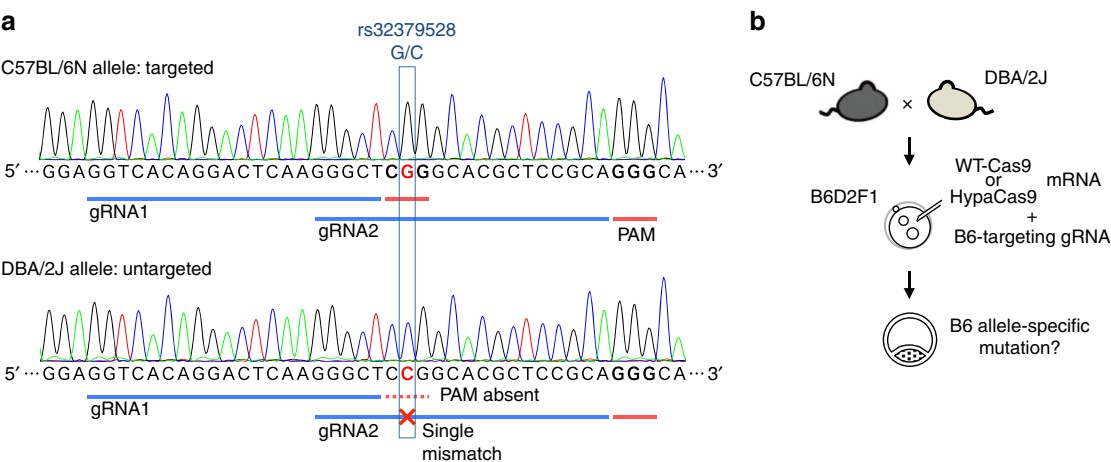

**Fig. 2** HypaCas9 can introduce allele-specific mutation by SNP discrimination. **a** Design of two C57BL/6N allele-targeting gRNAs in *Cdx2* exon 3. SNP rs32379528 exists in the PAM of gRNA1 and the protospacer of gRNA2. **b** Schematic for introducing allele-specific mutation in the B6D2F1 zygote by SNP discrimination. The results are shown in Table 1

| Table 1 The results of B6 allele-targeting mutation introduction by WT-Cas9 or HypaCas9 | | | | | |
|---|---|---|---|---|---|
| **Cross** | **Injected RNA** | **Embryos injected\*** | **Mutated** | | **Nonmutated (%)** |
| | | | **Biallelic (%)** | **Monoallelic (B6-specific) (%)** | |
| ♀B6 × ♂B6 | WT-Cas9 + gRNA1 | 19 | 15 (78.9) | 0 (0) | 4 (21.1) |
| ♀B6 × ♂D2 | WT-Cas9 + gRNA1 | 22 | 0 (0) | 9 (40.9) | 13 (59.1) |
| ♀B6 × ♂D2 | WT-Cas9 + gRNA2 | 21 | 21 (100) | 0 (0) | 0 (0) |
| ♀B6 × ♂D2 | HypaCas9 + gRNA2 | 23 | 0 (0) | 23 (100) | 0 (0) |
| *The total number of embryos in two replicate experiments is shown | | | | | |

HypaCas9 is expected to be useful for efficient generation of genetically modified viable pups and mouse strains.

In the present study, we utilized *Cdk1* as the target gene because it is known that *Cdk1* knockout mice exhibit embryonic lethality[27,28]. We aimed for B6 allele-specific deletion of *Cdk1* exon 3 by targeting two SNPs, rs29378304 and rs29355421, in the *Cdk1* intron (Fig. 3a). HypaCas9 mRNA and B6-targeting gRNAs were injected into B6D2F1 zygotes, and the embryos were transferred into oviducts of pseudopregnant foster mothers; 35 pups were obtained from 120 transferred embryos, and 22 of the pups had deletion in the B6 allele (Fig. 3b, c and Supplementary

Figs. 9 and 11C). These pups showed no morphological abnormality, indicating that they retained an intact allele (Fig. 3d). To confirm the heritability of the mutated allele, we generated F1 embryos by mating the mutated #6 F0 female mouse and WT male mouse. The target loci were sequenced, and we found that the deletion allele could be inherited in the next generation (Fig. 3e, f and Supplementary Fig. 11D). The inheritance of an unintended small indel mutation ($-5 + 3$) was also detected, suggesting the mosaic mutation in the #6 F0 mouse (Fig. 3f). These findings demonstrate that HypaCas9 is useful for the efficient generation of monoallelic genome-modified mice.

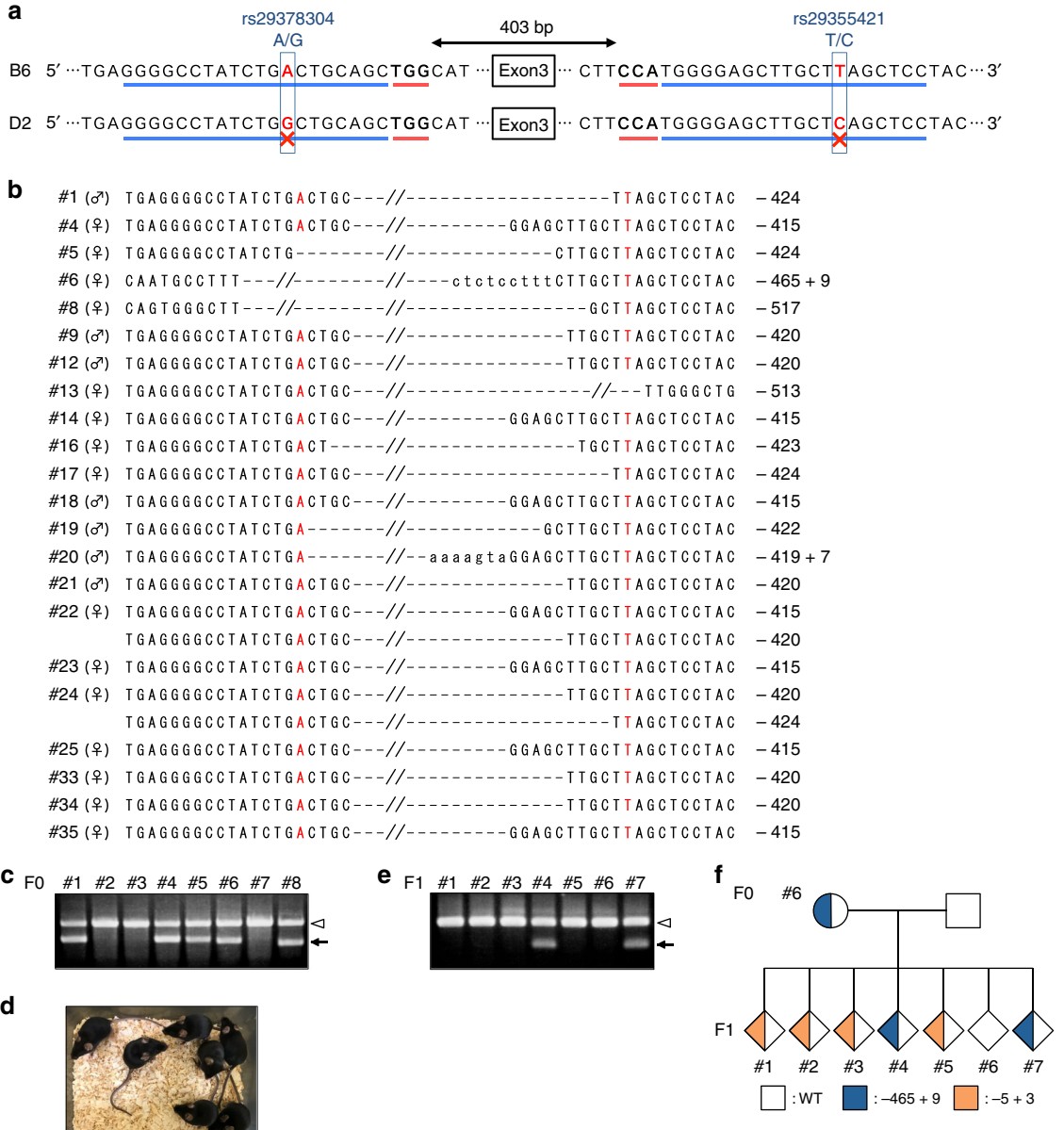

**Fig. 3** HypaCas9 enables efficient generation of monoallelic genome-modified mice. **a** Design of two gRNAs for B6 allele-specific deletion of *Cdk1* exon 3. The blue line indicates the gRNA target site and the red line indicates the PAM of gRNA. SNP rs29378304 is located in intron 2 of *Cdk1* and rs29355421 in intron 3. B6-targeting gRNAs were designed to have these SNPs within the protospacer. **b** Sequencing analysis of the deletion allele in F0 mice. The SNPs rs29378304 and rs29355421 are shown in red. **c** Representative PCR analysis of the *Cdk1* exon 3 deletion in F0 mice. **d** Representative F0 mice including both mice with and without the deletion allele. **e** PCR analysis of the *Cdk1* exon 3 deletion in F1 embryos, which were obtained by mating a #6 F0 female mouse with a WT male mouse. In **c** and **e**, the white arrowhead indicates the WT allele or an allele with a small indel. The arrow indicates the deletion allele. **f** Pedigree showing inheritance of the deletion allele of the #6 F0 mouse indicated in blue. Orange indicates the minor allele with a small indel ($-5 + 3$) caused by mosaic mutation, which is undetectable by sequencing in the #6 F0 mouse

## Discussion

In the present study, we attempted to apply the accuracy of HypaCas9 to genome editing in mouse zygotes. The results showed that HypaCas9 can discriminate even a single mismatch and ensured the off-target risk in zygotes was minimized. This accurate targeting can be applied to generate allele-specific genetically modified mice.

Conventionally used WT-Cas9 could not identify nucleotide mismatches between the target DNA sequence and DNA recognition sequence of gRNA in some cases, resulting in off-target mutations[5,11,12]. By contrast, HypaCas9 has enhanced target specificity and enables genetic modification without mutating similar sequences in other loci. In this study, HypaCas9 mutated the on-target locus in all the embryos analyzed, and exhibited reduced off-target activity with Gt(ROSA)26Sor-gRNA and Tyr-gRNA (Fig. 1 and Supplementary Fig. 4). Moreover, while WT-Cas9 has been reported to tolerate the PAM-distal mismatch[11,12], HypaCas9 showed mismatch intolerance regardless of the distance to the PAM in the mouse zygote, as similarly reported for human cells in culture (Fig. 1c and Supplementary Fig. 4D)[20]. However, further study at various loci is required to confirm the generality of accurate editing by HypaCas9, because it has been shown that the mutation efficiency is highly dependent on the locus and that high-fidelity Cas9 variants may exhibit reduced on-target activity depending on the locus[29]. Another interesting result shown in Fig. 1 is that WT-Cas9 mutated the off-target locus with mismatches at positions 5 and 17 in almost all the embryos (Fig. 1b), while no mutation was observed when using gRNA with a single mismatch at position 5 (Fig. 1c). It is possible that the probability of off-target mutation does not depend only on the number of mismatches, and that the difference in chromatin state between target loci influences the mutation efficiency. Furthermore, the results of this study suggest that the mosaic mutation can be caused by HypaCas9-mediated mutagenesis in zygotes (Fig. 3). In this study, we determined the existence of mutations by Sanger sequencing; therefore, there is a possibility that off-target mutations introduced in a small population of cells in mosaic embryos/pups might not be detected. More sensitive methods for mutation detection in whole cells of embryos and animals are expected to contribute to the rigorous evaluation of Cas9 accuracy in zygotes.

The accuracy of HypaCas9 would be effective for knock-in of a point mutation. Coinjection of ssODN with Cas9–gRNA can induce knock-in and nucleotide substitution at the target site[2,30,31], but the off-target effect of WT-Cas9 sometimes allows repeated targeting of the locus where the substitution has already been introduced, leading to unexpected mutations[25]. Although such re-editing can be avoided when the substituted nucleotide is designed to be located within PAM[25,26], this design limitation would be circumvented by the use of HypaCas9, due to its ability to discriminate a single mismatch in the protospacer of gRNA.

The targeting specificity of HypaCas9 could also be combined with a Cas9 variant having broad PAM compatibility. Recently, SpCas9-NG has been reported as a variant that recognizes NG PAMs instead of NGG and expands the targetable loci, and the addition of eSpCas9-derived alanine substitutions improved its targeting accuracy[32]. Therefore, the addition of HypaCas9-derived substitutions to SpCas9-NG would achieve even more accurate modification at target sites with NG PAMs.

Despite the finding that HypaCas9 precisely recognized target sequences in almost all of the loci, off-target mutation was detected in almost half of the embryos when using Gt(ROSA)26Sor-gRNA with a single mismatch at position 10 (mm10) (Fig. 1c), suggesting that HypaCas9 would require further improvement for accurate targeting. A Cas9 variant with a combination of alanine substitutions in eSpCas9 and SpCas9-HF1

exhibited improved specificity of target recognition[21]. Accordingly, it may be possible to improve the accuracy of HypaCas9 by addition of amino-acid substitutions derived from other high-fidelity Cas9 variants. As another strategy, the use of truncated gRNA (tru-gRNA) would be effective. tru-gRNA, whose protospacer length is truncated to <20 nucleotides, has been reported to enhance Cas9 specificity[33]. It can be expected that a combination of HypaCas9 with tru-gRNA will minimize the off-target effects.

Although multiple backcrossing is required in the case that a pure inbred background is necessary, SNP-mediated allele-specific modification using HypaCas9 would accelerate the analysis of genes essential for survival. In a conventional system, WT-Cas9 could induce biallelic mutation and generate knockout mice, but it is not useful for the generation of genetically modified viable pups and mouse strains when the target gene is essential for embryo development. Here, we have successfully generated monoallelic modification in the Cdk1 locus by distinguishing strain-specific SNPs, and the pups obtained—in which the mutated allele and the intact allele were detected—showed no abnormality (Fig. 3d). Because Cdk1 knockout is known to be embryonically lethal in mice[27,28], HypaCas9-mediated genetic modification in the mouse zygote would appear to be available for the efficient generation of essential gene-modified mouse strains. Furthermore, this high specificity of HypaCas9 for targeting alleles is also expected to facilitate allele-specific epigenome editing when combined with an endonuclease-dead Cas9 that is connected to the effector domains that modify the chromatin status[34,35].

## Methods

**Animals.** A 3-week-old female and >8-week-old male C57BL/6Ncr mice (Sankyo Labo Service Corporation, Tokyo), >8-week-old male DBA/2J mice (CLEA Japan, Tokyo), and 8-week-old female ICR mice (Sankyo Labo Service Corporation) were housed at $24 \pm 2\,°C$ and $50 \pm 10\%$ humidity under a 12/12 h light/dark cycle with free access to water and diet. All animal care and experiments conformed to the Guidelines for Animal Experiments of the University of Tokyo and were approved by the Animal Research Committee of the University of Tokyo (Approval No. P18-093).

**Construction of HypaCas9- and gRNA-plasmid DNA.** The HypaCas9 expression plasmid was constructed by mutation PCR using the pCAG-T3-hCAS9-pA plasmid vector (Addgene plasmid 48625)[5] to create N692A, M694A, Q695A, and H698A substitutions. The vector was sequenced using a commercial sequencing kit (Applied Biosystems, Foster City, CA) and a DNA sequencer (Applied Biosystems) according to the manufacturer's instructions. The plasmid vectors encoding each gRNA with T3 promoter were synthesized as described in a previous study[5]. We also used gRNA for the Gt(ROSA)26Sor locus, which has also been targeted in a previous study[5]. These vectors were sequenced as described above. All the sequence information is shown in Supplementary Fig. 10.

**DNA transfection and immunoblotting.** WT-Cas9 or HypaCas9 plasmid DNA was transfected into HEK293 cells using Lipofectamine LTX (Life Technologies, Carlsbad, CA) according to a previous report[36]. The transfected cells were harvested 20 h after transfection, and suspended in Laemmli buffer. Western blot analysis was performed according to the process described in our previous report[37]. The signals were detected using an Immunostar LD Kit (Wako, Tokyo) and a C-DiGit scanner (LI-COR, Lincoln, NE) (Supplementary Fig. 1).

**Transcription of Cas9 mRNA and gRNAs in vitro.** The Cas9 mRNA and gRNA were synthesized in vitro as described previously[5]. The RNA transcripts were precipitated with absolute ethanol, washed, and resuspended in RNase-free water (Gibco, Grand Island, NY). The RNA solutions were stored at $-80\,°C$ until use.

**Preparation of zygotes.** Mouse zygotes for microinjection were prepared by natural mating or in vitro fertilization. Sexually immature female C57BL/6NCr mice (3 weeks old) were superovulated by an intraperitoneal injection of 7.5 IU equine chorionic gonadotropin (eCG; ASKA Animal Health), followed by 7.5 IU human chorionic gonadotropin (hCG; ASKA Animal Health) 48 h later. These female mice were mated naturally with male DBA/2J mice (>8 weeks old) overnight, and zygotes were collected by oviductal flushing 20 h after hCG injection. For in vitro fertilization, cumulus-oocyte complexes were collected from oviductal

ampulla by oviductal flushing 14 h after hCG injection, and were cocultured with sperm collected from the cauda epididymidis of mature male C57BL/6NCr (>9 weeks old) or DBA/2J (>8 weeks old) mice in HTF medium. Pronuclei-formed zygotes by natural mating or IVF were transferred to M2 medium before microinjection.

**Microinjection and embryo transfer**. The microinjections were performed using a microscope equipped with a microinjector (Narishige, Tokyo). Approximately 4 pl of RNA solution, containing 10 ng μl$^{-1}$ of WT-Cas9 or HypaCas9 mRNA and 10 ng μl$^{-1}$ of gRNA (with 100 ng μl$^{-1}$ of eGFP mRNA only when the embryos were subjected to genotyping at the blastocyst stage), was injected into the cytoplasm of IVF-derived (Figs. 1 and 2) or natural mating-derived (Fig. 3) zygotes using continuous pneumatic pressure. The RNA-injected zygotes were cultured in KSOMaa-BSA, and after 24 h, two-cell embryos were transferred into the oviduct of pseudopregnant ICR female mice 0.5 days post coitum; alternatively, after 96 h, blastocysts were subjected to genotyping.

**Detection of induced mutations**. The genomic DNA was extracted from the blastocyst embryos or the tail tips of the pups according to our previous report[37], then subjected to PCR of the on-target locus and off-target loci using the primer sets shown in Supplementary Table 1. In the allele-specific gene modification using gRNA1 Cdx2 and gRNA2 Cdx2, PCR amplicons were digested using HindIII overnight. The PCR amplicons or digested products were purified by agarose gel electrophoresis, and extracted fragments were sequenced directly by Sanger sequencing using a commercial sequencing kit (Applied Biosystems) and a DNA sequencer (Applied Biosystems) according to the manufacturer's instructions. Each sequence chromatogram was analyzed with tracking of indels by decomposition (TIDE)[38] and mutation introduction was detected as insertion or deletion of nucleotides into the wild-type sequence or as overlapping of multiple chromatograms at the target site. The decomposition window was set to cover the largest possible window with high-quality traces, and when the indel frequency was higher than 10%, the sample was determined to be mutated. In Fig. 2, the sequencing data from HindIII-digested products were analyzed using CRISP-ID[39] because of its low R-squared value in TIDE analysis.

**Reproducibility**. Two or more replicates were performed in all experiments. Zygotes used for microinjections were randomly picked up from in vitro or naturally fertilized zygotes pools. In the genotyping at the blastocyst stage, the embryos were randomly chosen from the microinjection-derived GFP-positive blastocysts.

**Reporting summary**. Further information on research design is available in the Nature Research Reporting Summary linked to this article.

## Data availability

The sequence information of the plasmids is described in the Supplementary Information and plasmid DNAs have deposited in Addgene (#131467, https://www.addgene.org/Wataru_Fujii/). The data that support the findings of this study are available on request from the corresponding author.

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

## Acknowledgements

We thank members of the Applied Genetics laboratory for their assistance with the study. This work was supported by Grant-in-Aid for Scientific Research [26712025 and 18K19261 to W.F.] and a Grant-in-Aid for JSPS Research Fellow [19J22776 to A.I.] from the Japan Society for the Promotion of Science.

## Author contributions

A.I. and W.F. conceived, designed and performed the experiments, and analyzed the data. A.I., W.F., K.S., and K.N. were contributed reagents/materials/analysis tools. A.I. and W.F. wrote the paper. All authors read and approved the final paper.

## Competing interests

The authors declare no competing interests.
