## [Peer Review File · Communications Biology]

Reviewers' comments:

Reviewer #1 (Remarks to the Author):

Summary:

The repurposing of CRISPR nucleases for genome editing has revolutionized biomedical research, including the pace and efficacy of animal model generation. One major concern with genome editing nucleases (including the commonly used SpCas9) is their propensity to recognize and cleave off-target sites that are similar in sequence to the intended on-target site. To circumvent this potential issue, several research groups have developed engineered CRISPR nucleases that have improved specificity profiles. The present work seeks to evaluate the on- and off-target activity of one of the improved specificity nucleases, HypaCas9, in the context of mouse zygote injections.

While off-target concerns are problematic for certain genome editing applications, they are less of a concern for animal and crop generation since these organisms can generally be bred through subsequent generations to passage out potential off-target events. That being said, should an enhanced specificity Cas9 variant perform optimally in mouse zygotes, this would circumvent the necessity of successive breeding cycles.

Towards understanding the on-target activity and specificity of HypaCas9 in mouse zygotes, the authors compare the activities of wild-type SpCas9 and HypaCas9 in several formats (including for allele specificity). While the authors observe robust on-target editing (and in most cases no off-target editing), the number of target sites examined is low in number, preventing the authors from confidently making general statements. Furthermore, there is insufficient detail for the readership to understand how the authors are analyzing on- and off-target editing events. For the manuscript to be acceptable for publication, the authors should: (1) sufficiently caveat their statements given their small sample sizes, (2) provide additional details about how they evaluated editing, and (3) perform substantial text edits to remove typos, grammatical errors, and improve the clarity of their narrative.

Major Comments:

1. In the absence of the authors performing additional experiments at alternate loci, the language about the generalizability of using HypaCas9 should be tempered. Simply because HypaCas9 worked exceptionally well in terms of on-target editing and specificity at these few loci, does not mean that these are generalizable observations. For example, the statement on lines 57-59 is far too general given the limited sampling performed by the authors. The authors should add caveats to this and similar statements, given that they examined specificity at so few target sites.
2. It is not clear how the authors quantified on- or off-target events. In the methods section, the authors write that: "The predictive mutation pattern of each allele was analyzed by sequence data", which is an unacceptable level of detail for the readership. The authors should clarify whether the PCR products were sequenced by Sanger or NGS, and by which method the data was quantified.
3. The authors should also make their data available through the NCBI SRA or another equivalent data repository.
4. What level of on- or off-target editing are the authors considering significant enough to warrant being called as an event in Figs. 1c and 1d? There is an insufficient level of detail surrounding how the authors are determining on- and off-target editing events. The authors state that "HypaCas9 showed no mutation introduction in the potential off-target site while retaining on-target activity comparable to WT-Cas9.", yet this claim is not adequately supported (eg., what does 'comparable' mean?). The authors should include a table with the quantified levels of on- and off-target editing for all experiments.
5. The manuscript would benefit substantially from English language editing, as there are that necessitate substantial revisions. This reviewer started to indicate several instances in the minor comments below, however, this is not a comprehensive list.

Minor Critiques:

1. On line 16 of page 1, the phrase 'expanded the targetable loci' isn't technically true, since HypaCas9 has the targeting range as SpCas9 (since both proteins recognize an NGG PAM).
2. On lines 19/20, sgRNA stands for "single guide RNA", not "synthetic guide RNA".
3. On line 21, "indels" is not defined.
4. On line 25, the sentence "Cas9 nickase and FokI-dCas9 recognize the target locus more precisely and reduce off-target effects" is not technically true. Cas9 nickase and dCas9 have the same apparent target specificity as Cas9 – they improve specificity by other mechanisms.
5. Figure 1A isn't necessary and can simply be described as the authors have done in the text.
6. On lines 64/65, the authors should cite examples from the literature (from 5+ years ago) that showed evidence of Cas9 being unable to distinguish targets that differ by single nucleotides. This is a well-established phenomena.
7. It would be helpful for the authors to annotate different sequence elements in the sequences of Supplementary Figure S1 (by different colors / bold / italics / underline) to differentiate important sequence features (NLS, promoter, etc.)

Reviewer #2 (Remarks to the Author):

Comments to the authors

COMMSBIO-19-1528

Title: Accuracy of HypaCas9 facilitates allele-specific gene modification in mouse zygotes

The authors of the manuscript have described their work on the use of a previously generated high fidelity version of Cas9, namely HypaCas9, to reduce off-target mutations during mouse transgenesis. Furthermore, they convincingly present a prove-of-principle approach for mono-allelic mutation of an essential gene in vivo by means of HypaCas9. The study is technically sound, original and the presented results will be of relevance to the broad community of researchers involved in CRISPR/Cas9 mediated transgenesis. Only the lack of additional experiments to support the conclusion of reduced off-target activity of HypaCas9 in vivo as well as some more elaborate discussion of the literature on high fidelity Cas9 versions hinder the immediate publication. In addition, some paragraphs may benefit from re-phrasing to better convey the relevant findings and conclusions, respectively. A revised version of the manuscript addressing these issues will in my opinion add to the knowledge base and will be worth publishing.

Major points:

- The conclusion of lower off-target and unaffected on-target editing by HypaCas9 compared to WT-Cas9 in vivo is based on transgenesis with only one gRNA. Many previous publications of high fidelity Cas9 variants in vitro and even one in vivo study have demonstrated that editing specificity and efficiency is highly dependent on the locus and the gRNA, respectively (e.g.: doi.org/10.1038/s41591-018-0137-0, doi.org/10.1038/s41467-018-05477-x, doi.org/10.1038/s41592-018-0011-5). I therefore strongly suggest to include data for off- and on-target editing of at least another locus/gRNA in the manuscript or the supplement (i.e. repeat fig 1C using another gRNA). HypaCas9 and virtually all other high fidelity Cas9 variants have been shown to exhibit reduced on-target efficiency on a number of loci (same literature as above). I would therefore like to encourage the authors to include this issue in the discussion of the manuscript.

- Seven high fidelity Cas9 have been reported to date. HiFi Cas9, eSpCas9(1.1), SpCas9-HF1, evoCas9, Sniper-Cas9, HypaCas9 and HeFSpCas9 variants. The authors cite only three in the introduction and an additional one in the discussion. It would improve the manuscript in my opinion to cite all available high fidelity Cas9 variants or better even include them in an additional paragraph of the discussion (doi.org/10.1038/s41591-018-0137-0 includes all references for the additional variants but the publication on the HeFSpCas9 variants which has been cited by the authors already).
- As many of the quantifications in the manuscript represent Sanger sequencing data (e.g fig1CD) I would suggest to include at least some representative chromatograms of these in the supplement of the manuscript. It will be most convincing for the reader to see a representative part of the raw data.

Minor points

Chronological:

- 20-22: The sentence implies that double-strand DNA breaks always result in bi-allelic INDELS. However, double strand breaks (i.e. by Cas9 nuclease) may result in INDELS but are mostly repaired without any INDELS. In addition, Cas9 activity does by far not result only in bi-allelic modifications but also in mono-allelic. Please re-phrase the sentence.
- 24-27: The sentence implies that FokI-dCas9 approaches need two gRNAs. However, this limitation only counts for the also mentioned Cas9 nickase approach. Please re-phrase the sentence.
- 30-33: I would suggest to attenuate the claim regarding the lowest off-target efficiency of HypaCas9 compared to other available high fidelity Cas9 variants. For example Sniper-Cas9 has been shown to exhibit even lower off-target efficiency than HypaCas9 in vitro (doi.org/10.1038/s41467-018-05477-x).
- 43 – 45: It is not clear to me, if the expression vector backbone alone or the HypaCas9 sequence has been altered ('optimized'). This should be clearly stated.
- 47: Please use the standard nomenclature for genetic loci in the entire manuscript (i.e. "Gt(ROSA)26Sor" instead of "Rosa26")
- 48: Please mention in either the text or the caption which method has been used (i.e. Sanger sequencing).
- 56: It is highly interesting that WT-Cas9 did edit the analyzed off-target with mismatches at position 5 and 17 in virtually all embryos (fig1C, 3rd column) whereas it did not edit any embryo when the gRNA was mutated at position 5 only (fig1D, 3rd column). Could the authors speculate about that in the discussion? Is it maybe a result of the nature of the mismatch (i.e. on the DNA vs gRNA)?
- 71: The idea of separating the inherited allele from the two substrains by RFLP is most interesting but new to me. It would be most informative to the reader if this was mentioned in the result part and not only visible in the supplement (i.g. figS3).
- 74-75: It is a very impressive result that HypaCas9 can be used to introduce allele specific mutations in such a highly specific manner (fig2C, line 3 vs. 4)! It may be worth highlighting this result even more in the result section.
- 79: The applied approach is most useful for essential genes. However, the embryonic lethality of the utilized gene (Cdk1) is only mentioned very late in the discussion (lines 134/135). I would suggest to emphasize this in the result section.
- 78-87: The manuscript may benefit a lot from additional explanation of the results and modifications of the corresponding fig3.
 - o It is a bit unfortunate that the 5' sequence of the most important founder (#6) is not displayed (fig3B).
 - o The result of fig3C (no abnormalities in the litter) is not mentioned in the entire paragraph/result section but only in the discussion.
 - o Fig3D may benefit from indicating that the results shown from the F1 generation is the offspring of founder #6.

o Fig3E is very difficult to understand due to the additional finding of unintended INDELS (-5+3). The authors may not only explain this result in more details but also include it in the discussion (is it due to mosaics that they did not find these small INDELS during the sequencing?).

- 93-104: The phrasing of the paragraph is not easy to understand. Re-editing is often tried to be prevented by mutating the PAM site in the gRNA. My interpretation is, that could be avoided by use of HypaCas9.

- 121-127: The authors compare their results of mono-allelic editing with "zygote-optimized" and "highly active" HypaCas9 to mono-allelic editing in rat embryos with WT-Cas9 in a previous publication [28] and attribute the difference to lower Cas9 nuclease activity of the employed WT-Cas9. How exactly the HypaCas9 has been optimized for zygote expression (see comment to lines 43-45) or the attributed enhanced nuclease activity, and in which way this contributes to the higher specificity of HypaCas9 in general is not well explained. The manuscript may benefit from re-phrasing of even removal of this paragraph.

- 127-134: The authors may want to re-phrase the paragraph as it may implicate that the use of WT-Cas9 predominantly or always leads to biallelic modification, which is, according to own experience and published literature, not the case.

General: the authors may want to include a brief sentence that editing of hybrid zygotes in their approach may raise the need for multiple back-crossing of the founder mice to C57BL/6N mice in order to generate a pure inbred background needed for many scientific questions.

- 142-144: Please add to the ethic statement:

- o The information to mice vendors as given in the 'reporting summary', and the facility where the mice has been housed or the research has been carried out.

- o Fundamentals of animal care and housing and hygiene status

- 168: please indicate the vendor of the hormones

- 191: Please indicate the method of sequencing (I guess Sanger sequencing). It is unfortunately not mentioned anywhere in the manuscript.

- 195: Change to the adequate location at Nature Communications Biology online

RESPONSES TO REFEREE #1's COMMENTS:

Thank you very much for your careful review of our manuscript and giving us your helpful advices and suggestions. Based on your comments, we have made changes to our manuscript as follows.

Major Comments:

1. COMMENT: In the absence of the authors performing additional experiments at alternate loci, the language about the generalizability of using HypaCas9 should be tempered. Simply because HypaCas9 worked exceptionally well in terms of on-target editing and specificity at these few loci, does not mean that these are generalizable observations. For example, the statement on lines 57-59 is far too general given the limited sampling performed by the authors. The authors should add caveats to this and similar statements, given that they examined specificity at so few target sites.

RESPONSE: To validate the accuracy of HypaCas9 at another locus, we have performed the additional experiments at *Tyrosinase (Tyr)* locus in the same manner as done at Gt(ROSA)26Sor in Figure 1, and the results were added to the revised manuscript on lines 69–78, and to Supplementary Figure S4. This additional data supported the improved fidelity of HypaCas9 in mouse zygotes, but we agree with your suggestion that these results are insufficient to be generalized. Therefore, we have tempered the explanation in the result section and added the following caveat against the generalizability of using HypaCas9 to the discussion section (lines 139–143):

“However, further study at various loci is required to confirm the generality of accurate editing by HypaCas9, because it has been shown that the mutation efficiency is highly dependent on the locus and that high-fidelity Cas9 variants may exhibit reduced on-target activity depending on the locus [29].”

2. COMMENT: It is not clear how the authors quantified on- or off-target events. In the methods section, the authors write that: “ The predictive mutation pattern of each allele was analyzed by sequence data”, which is an unacceptable level of detail for the readership. The authors should clarify whether the PCR products

were sequenced by Sanger or NGS, and by which method the data was quantified.

RESPONSE: PCR products in all experiments were sequenced by Sanger sequencing, and whether mutation was introduced in each embryo was determined based on the chromatogram. To clarify the procedure, we have added the explanations to the result section (lines 56–57), and revised “Detection of induced mutations” in materials and methods section (lines 252–257).

3. COMMENT: The authors should also make their data available through the NCBI SRA or another equivalent data repository.

RESPONSE: As described above, mutation introduction was assessed by Sanger sequencing. To provide the sequencing data from which mutation introduction was detected, we have added the representative sequencing chromatograms in all experiments to Supplementary Figure S2, 3, 5, 6, and 8.

4. COMMENT: What level of on- or off-target editing are the authors considering significant enough to warrant being called as an event in Figs. 1c and 1d? There is an insufficient level of detail surrounding how the authors are determining on- and off-target editing events. The authors state that “HypaCas9 showed no mutation introduction in the potential off-target site while retaining on-target activity comparable to WT-Cas9.”, yet this claim is not adequately supported (eg., what does ‘comparable’ mean?). The authors should include a table with the quantified levels of on- and off-target editing for all experiments.

RESPONSE: Whether Cas9–gRNA introduced mutation into the on- and off-target locus was assessed by Sanger sequencing in each embryo, and the incidences of target editing in each experiment are indicated by the number of embryos in which the mutation introduction was detected. Our results showed that the incidence of embryos in which off-target mutation was introduced was decreased by using HypaCas9 compared to WT-Cas9 in all experiments. However, we did not carry out statistic analysis because we were unable to prepare a sufficient number of embryos for statistics, and therefore we revised this sentence to temper the notation about quantitativity (lines 58–62).

5. COMMENT: The manuscript would benefit substantially from English language editing, as there are that necessitate substantial revisions. This reviewer started to indicate several instances in the minor comments below, however, this is not a comprehensive list.

RESPONSE: We apologize for a large number of English mistakes and appreciate for your kind pointing out. The manuscript was proofread by a native speaker of English from a company providing professional English editing services, but as you suggested, there still remained some mistakes and insufficient explanations. We have revised the manuscript in line with the comments you and another reviewer kindly pointed out, and the revised manuscript has been proofread by another English editing service.

Minor Critiques:

1. COMMENT: On line 16 of page 1, the phrase ‘expanded the targetable loci’ isn’t technically true, since HypaCas9 has the targeting range as SpCas9 (since both proteins recognize an NGG PAM).

RESPONSE: Thank you for drawing this problem to our attention. The revised version reads as follows (lines 16–17):

“the improved accuracy of HypaCas9 has expanded the range of applicable loci for the efficient generation of genetically modified animals.”

2. COMMENT: On lines 19/20, sgRNA stands for “single guide RNA”, not “synthetic guide RNA”.

RESPONSE: Thank you for pointing out this error. It should be “single guide RNA”. This error has been corrected in the revised manuscript (line21).

3. COMMENT: On line 21, “indels” is not defined.

RESPONSE: We have added “insertions and deletions” as the definition of “indels” (lines 23–24).

4. COMMENT: On line 25, the sentence “Cas9 nickase and FokI-dCas9 recognize the target locus more precisely and reduce off-target effects” is not technically true. Cas9 nickase and dCas9 have the same apparent target specificity as Cas9 – they improve specificity by other mechanisms.

RESPONSE: As pointed out, we incorrectly stated about the mechanism of specificity of Cas9 nickase and FokI-dCas9. The sentence has been revised as follows (lines 28–29):

“It has been reported that the use of paired Cas9 nickase or FokI-dCas9 can reduce the off-target effect”

5. COMMENT: Figure 1A isn’t necessary and can simply be described as the authors have done in the text.

RESPONSE: We have deleted Figure 1A and added the simplified version to Supplementary Figure S1A.

6. COMMENT: On lines 64/65, the authors should cite examples from the literature (from 5+ years ago) that showed evidence of Cas9 being unable to distinguish targets that differ by single nucleotides. This is a well-established phenomena.

RESPONSE: We agree that off-target mutation into the single-nucleotide mismatched sequence is a well-established phenomenon. We have cited an additional reference and revised the sentence as follows (lines 86–89):

“By contrast, when a mismatched nucleotide is positioned in the protospacer of gRNA, the off-target effect of WT-Cas9 can introduce mutation into an untargeted allele with a single mismatch, as indicated in previous studies [11,12] and by the results in Fig.1.”

7. COMMENT: It would be helpful for the authors to annotate different sequence elements in the sequences of Supplementary Figure S1 (by different colors / bold / italics / underline) to differentiate important sequence features (NLS, promoter, etc.)

RESPONSE: Thank you for this suggestion. We have annotated important different sequence features by different colors in the sequences of Supplementary Figure S10.

RESPONSES TO REFEREE #2's COMMENTS:

Thank you very much for your careful review of our manuscript and giving us your helpful advices and suggestions. Based on your comments, we have made changes to our manuscript as follows.

Major Comments:

1. COMMENT: The conclusion of lower off-target and unaffected on-target editing by HypaCas9 compared to WT-Cas9 in vivo is based on transgenesis with only one gRNA. Many previous publication of high fidelity Cas9 variants in vitro and even one in vivo study have demonstrated that editing specificity and efficiency is highly dependent on the locus and the gRNA, respectively (e.g.: doi.org/10.1038/s41591-018-0137-0, doi.org/10.1038/s41467-018-05477-x, doi.org/10.1038/s41592-018-0011-5). I therefore strongly suggest to include data for off- and on-target editing of at least another locus/gRNA in the manuscript or the supplement (i.e. repeat fig 1C using another gRNA). HypaCas9 and virtually all other high fidelity Cas9 variants have been shown to exhibit reduced on-target efficiency on a number of loci (same literature as above). I would therefore like to encourage the authors to include this issue in the discussion of the manuscript.

RESPONSE: According to your suggestion, we have performed the two additional experiments at *Tyrosinase* (*Tyr*) locus. As with Gt(ROSA)26Sor-gRNA in Figure 1, HypaCas9 showed the reduced off-target mutation rates with gRNAs designed at *Tyr* locus. These results were added to the revised manuscript on lines 69–78, and to Supplementary Figure S4.

In our experiments, HypaCas9 could introduce mutation at on-target locus as efficiently as WT-Cas9. However, HypaCas9 may have lower activity compared to WT-Cas9 in mammalian zygotes as shown in the previous reports that you presented. Therefore, we have added the following explanation to the discussion section (lines 140–143):

“However, further study at various loci is required to confirm the generality of accurate editing by HypaCas9, because it has been shown that the mutation efficiency is highly dependent on the locus and that high-fidelity Cas9 variants may exhibit reduced on-target activity depending on the locus [29].”

2. COMMENT: Seven high fidelity Cas9 have been reported to date. HiFi Cas9, eSpCas9(1.1), SpCas9-HF1, evoCas9, Sniper-Cas9, HypaCas9 and HeFSpCas9 variants. The authors cite only three in the introduction and an additional one in the discussion. It would improve the manuscript in my opinion to cite all available high fidelity Cas9 variants or better even include them in an additional paragraph of the discussion (doi.org/10.1038/s41591-018-0137-0 includes all references for the additional variants but the publication on the HeFSpCas9 variants which has been cited by the authors already).

RESPONSE: We appreciate you pointing out that we have missed some of the variants. We have added four references to the introduction (line 34).

3. COMMENT: As many of the quantifications in the manuscript represent Sanger sequencing data (e.g fig1CD), I would suggest to include at least some representative chromatograms of these in the supplement of the manuscript. It will be most convincing for the reader to see a representative part of the raw data.

RESPONSE: According to your suggestion, we have added the representative sequencing chromatograms of mutated and non-mutated embryos to Supplementary Figure S2, 3, 5, 6, and 8.

Minor points

Chronological:

1. COMMENT: 20-22: The sentence implies that double-strand DNA breaks always result in bi-allelic INDELS. However, double strand breaks (i.e. by Cas9 nuclease) may result in INDELS but are mostly repaired without any INDELS. In addition, Cas9 activity does by far not result only in bi-allelic modifications but also in mono-allelic. Please re-phrase the sentence.

RESPONSE: We agree that double strand breaks do not always result in bi-allelic INDELS, and so we have deleted the words “of both alleles” (line 24).

2. COMMENT: 24-27: The sentence implies that FokI-dCas9 approaches need two gRNAs. However, this limitation only counts for the also mentioned Cas9 nickase approach. Please re-phrase the sentence.

RESPONSE: As shown in the previous studies (ref 16; Wyvekens et al., 2015; Terao et al., 2016), FokI domain of FokI-dCas9 requires dimerization for its nuclease activity. Each FokI-dCas9 requires one gRNA, and therefore the gene editing by dimerized FokI-dCas9 needs two gRNAs to be designed at target locus in the same way as the double-nicking approach using Cas9 nickase.

3. COMMENT: 30-33: I would suggest to attenuate the claim regarding the lowest off-target efficiency of HypaCas9 compared to other available high fidelity Cas9 variants. For example Sniper-Cas9 has been shown to exhibit even lower off-target efficiency than HypaCas9 in vitro (doi.org/10.1038/s41467-018-05477-x).

RESPONSE: According to your suggestion, this sentence has been changed as follows (lines 34–35):

“One of these variants, hyper-accurate (Hypa) Cas9, exhibited significantly higher accuracy than WT-Cas9 and minimized off-target cleavage [20].”

4. COMMENT: 43 – 45: It is not clear to me, if the expression vector backbone alone or the HypaCas9 sequence has been altered (‘optimized’). This should be clearly stated.

RESPONSE: We apologized for the insufficient description. To improve the clarity, the sentence has been changed as follows (lines 49–52):

“We constructed a HypaCas9 vector by introducing the alanine substitutions into the previously reported WT-Cas9 vector, which is optimized for expression in mouse zygotes by addition of the 3’UTR sequence of Tbp11 and a 95nt polyadenine tail sequence [5]. It was confirmed that the substitutions did not affect the level of protein expression (Supplementary Figure S1B).”

5. COMMENT: 47: Please use the standard nomenclature for genetic loci in the entire manuscript (i.e. “Gt(ROSA)26Sor” instead of “Rosa26”)

RESPONSE: According to your indication, we corrected this word in the revised manuscript.

6. COMMENT: 48: Please mention in either the text or the caption which method has been used (i.e. Sanger sequencing).

RESPONSE: According to your suggestion, we have added the phrase “by Sanger sequencing” to the manuscript in result section (line 57) and in method section (line 252).

7. COMMENT: 56: It is highly interesting that WT-Cas9 did edit the analyzed off-target with mismatches at position 5 and 17 in virtually all embryos (fig1C, 3rd column) whereas it did not edit any embryo when the gRNA was mutated at position 5 only (fig1D, 3rd column). Could the authors speculate about that in the discussion? Is it maybe a result of the nature of the mismatch (i.e. on the DNA vs gRNA)?

RESPONSE: Thank you for this suggestion. We agree that it is interesting point. At present, no definitive conclusion has been reached, but we can speculate that this difference in mutation efficiency is attributed to difference in the condition of each locus, such as chromatin state. Therefore, we added the following explanation to discussion section (lines 143-147):

“Another interesting result shown in Fig. 1 is that WT-Cas9 mutated the off-target locus with mismatches at positions 5 and 17 in almost all the embryos (Fig. 1B), while no mutation was observed when using gRNA with a single mismatch at position 5 (Fig. 1C). It is possible that the probability of off-target mutation does not depend only on the number of mismatches, and that the difference in chromatin state between target loci influences the mutation efficiency.”

8. COMMENT: 71: The idea of separating the inherited allele from the two substrains by RFLP is most interesting but new to me. It would be most informative to the reader if this was mentioned in the result part and not only visible in the supplement (i.g. figS3).

RESPONSE: We have added the following explanation about the method for allele separation to the result part (lines 95–98):

“The allele separation was carried out by PCR-RFLP using an SNP (NCBI dbSNP Build 142; rs29679715: B6, T/T; D2, A/A) (Supplementary Figure S7). HindIII cleaves only the D2 allele at the SNP locus, leading to the allele separation in electrophoresis.”

9. COMMENT: 74-75: It is a very impressive result that HypaCas9 can be used to introduce allele specific mutations in such a highly specific manner (fig2C, line 3 vs. 4)! It may be worth highlighting this result even more in the result section.

RESPONSE: Thank you for your comment. We have emphasized this result by addition of the following sentence (lines 103–104):

“HypaCas9 successfully discriminated the untargeted allele with single mismatch in a highly specific manner in contrast to WT-Cas9.”

10. COMMENT: 79: The applied approach is most useful for essential genes. However, the embryonic lethality of the utilized gene (Cdk1) is only mentioned very late in the discussion (lines 134/135). I would suggest to emphasize this in the result section.

RESPONSE: According to your suggestion, we have included the additional explanation about the applicability for essential genes in the result section as follows (lines 108–113):

“Finally, we applied HypaCas9 to generate allele-specific genome-modified mice targeting the gene essential for survival. When targeting such genes, in which biallelic mutation results in lethality, the deterministic monoallelic gene modification using HypaCas9 is expected to be useful for efficient generation of genetically modified viable pups and mouse strains.

In the present study, we utilized Cdk1 as the target gene because it is known that Cdk1 knockout mice exhibit embryonic lethality [27,28].”

11. COMMENT: 78-87: The manuscript may benefit a lot from additional explanation of the results and modifications of the corresponding fig3. It is a bit unfortunate that the 5' sequence of the most important founder (#6) is not displayed (fig3B).

RESPONSE: Thank you for pointing out this issue. In the revised version of Fig.3B, we have included the 5' sequence of deletion region in #6 and #8, and the 3' sequence in #13.

12. COMMENT: The result of fig3C (no abnormalities in the litter) is not mentioned in the entire paragraph/result section but only in the discussion.

RESPONSE: We have changed the sentence in lines 116–119 as follows:

“35 pups were obtained from 120 transferred embryos, and 22 of the pups had deletion in the B6 allele (Fig.3B and C and Supplementary Figure S9). These pups showed no morphological abnormality, indicating that they retained an intact allele (Fig.3D).”

13. COMMENT: Fig3D may benefit from indicating that the results shown from the F1 generation is the offspring of founder #6.

RESPONSE: We have changed the sentence in lines 119-121 as follows:

“To confirm the heritability of the mutated allele, we generated F1 embryos by mating the mutated #6 F0 female mouse and WT male mouse.”

14. COMMENT: Fig3E is very difficult to understand due to the additional finding of unintended INDELs (-5+3). The authors may not only explain this result in more details but also include it in the discussion (is it due to mosaics that they did not find these small INDELs during the sequencing?).

RESPONSE: We have added the following explanations about this indel to the result

section (lines 122–123):

“The inheritance of an unintended small indel mutation (-5+3) was also detected, suggesting the mosaic mutation in the #6 F0 mouse (Fig.3F).”

15. COMMENT: 93-104: The phrasing of the paragraph is not easy to understand. Re-editing is often tried to be prevented by mutating the PAM site in the gRNA. My interpretation is, that could be avoided by use of HypaCas9.

RESPONSE: We apologize for the confused description. We have divided this paragraph into two paragraphs (lines 131–156) and made some modifications to improve the clarity (lines 148–156).

16. COMMENT: 121-127: The authors compare their results of mono-allelic editing with “zygote-optimized” and “highly active” HypaCas9 to mono-allelic editing in rat embryos with WT-Cas9 in a previous publication [28] and attribute the difference to lower Cas9 nuclease activity of the employed WT-Cas9. How exactly the HypaCas9 has been optimized for zygote expression (see comment to lines 43-45) or the attributed enhanced nuclease activity, and in which way this contributes to the higher specificity of HypaCas9 in general is not well explained. The manuscript may benefit from re-phrasing of even removal of this paragraph.

RESPONSE: We have deleted these sentences.

17. COMMENT: 127-134: The authors may want to re-phrase the paragraph as it may implicate that the use of WT-Cas9 predominantly or always leads to biallelic modification, which is, according to own experience and published literature, not the case.

RESPONSE: We recognize that WT-Cas9 does not always introduce biallelic mutation as reported previously by some groups [Mashiko et al., Sci Rep, 2013; Yoshimi et al., Nat Commun, 2014], but it has also been reported that WT-Cas9 could introduce biallelic mutation in almost all mouse embryos by other groups including ours [Wang et al., Cell, 2013; Fujii et al., Nucleic Acids Res, 2013; Fujii et al., J Reprod Dev, 2014;

Horii et al., Sci Rep, 2014]. Therefore, we consider that the efficiencies of biallelic modification by WT-Cas9 vary between the systems employed, and the allele-specific gene modification established in this study would be effective in the highly efficient modification systems. However, to avoid misleading description, we have attenuated the explanation in this paragraph (lines 180–186).

18. COMMENT: General: the authors may want to include a brief sentence that editing of hybrid zygotes in their approach may raise the need for multiple back-crossing of the founder mice to C57BL/6N mice in order to generate a pure inbred background needed for many scientific questions.

RESPONSE: Backcrossing is certainly required in the experiments that need pure inbred mice, as you indicated. We have added the following explanation to the sentence in lines 180-181:

“Although multiple backcrossing is required in the case that a pure inbred background is necessary,”

19. COMMENT: 142-144: Please add to the ethic statement: The information to mice vendors as given in the ‘reporting summary’, and the facility where the mice has been housed or the research has been carried out. Fundamentals of animal care and housing and hygiene status.

RESPONSE: We have added the details of animals to this part as follows (lines 197–204):

“Animals

3-week-old female and >8-week-old male C57BL/6Ncr mice (Sankyo Labo Service Corporation, Tokyo), >8-week-old male DBA/2J mice (CLEA Japan, Tokyo), and 8-week-old female ICR mice (Sankyo Labo Service Corporation) were housed at $24 \pm 2^{\circ}\text{C}$ and $50 \pm 10\%$ humidity under a 12/12 h light/dark cycle with free access to water and diet. All animal care and experiments conformed to the Guidelines for Animal Experiments of The the University of Tokyo and were approved by the Animal Research Committee of The the University of Tokyo (Approval No. P18-093).”

20. COMMENT: 168: please indicate the vendor of the hormones.

RESPONSE: We have added the vendor of the hormones to the “preparation of zygotes” part (lines 229–230).

21. COMMENT: 191: Please indicate the method of sequencing (I guess Sanger sequencing). It is unfortunately not mentioned anywhere in the manuscript.

RESPONSE: To clarify the explanation about sequencing methods, we have revised the “detection of induced mutations” part in material and methods (lines 252–257).

22. COMMENT: 195: Change to the adequate location at Nature Communications Biology online.

RESPONSE: Thank you for pointing out this mistake. It has been corrected in the revised manuscript (line 260).

Reviewers' comments:

Reviewer #1 (Remarks to the Author):

The authors have made a good effort to address most of the requested revisions, adding new results at a second target site/genomic locus. This additional data support their initial conclusions. The main result of this manuscript is an interesting finding, however, the manuscript still remains subject to some caveats, including:

(1) Issues of sensitivity. The authors Sanger sequence the pups to genotype them, which potentially has issues with mosaic edited animals (especially given that the Cas9 proteins are delivered via mRNA rather than RNP). The authors should discuss this and the potential implications for their ability to detect off-targets in mosaic animals. One could imagine that less efficiently targeted off-target sites kinetically occur at later developmental stage (before the mRNA/nuclease is effectively diluted out), rendering it difficult to determine by Sanger sequencing whether or not an edit truly occurred.

(2) Related to point #1, the authors don't provide a metric as to how they determine whether a Sanger trace reveals evidence of editing or not. There are freely available websites/software that can predict editing results (%) based on Sanger trace deconvolution (TIDE, ICE, etc.). The authors should be more clear about their judgement calls and use a more rigorous metric to quantify edited vs. un-edited samples based on empirical evidence.

(3) The English language and grammar of the article remains a major issue and should be appropriately revised.

(4) The author's response to Minor Comment #1 still remains incorrect. The modified sentence: "the improved accuracy of HypaCas9 has expanded the range of applicable loci for the efficient generation of genetically modified animals." continues to imply that HypaCas9 EXPANDS the range of targetable loci, which simply isn't true. HypaCas9 is still subject to the same NGG PAM requirement as wild-type SpCas9, so the 'range of applicable loci' remain the same (and is not expanded).

Reviewer #2 (Remarks to the Author):

The authors fully addressed the concerns addressed in the review.

We can therefore recommend this article for publication without further requirements.

RESPONSES TO REFEREE #1's COMMENTS:

Thank you very much for your careful re-review of our revised manuscript (COMMSBIO-19-1528) and giving us your helpful advices and suggestions. Based on your comments, we have made changes to our manuscript as follows.

Major Comments:

1. COMMENT: Issues of sensitivity. The authors Sanger sequence the pups to genotype them, which potentially has issues with mosaic edited animals (especially given that the Cas9 proteins are delivered via mRNA rather than RNP). The authors should discuss this and the potential implications for their ability to detect off-targets in mosaic animals. One could imagine that less efficiently targeted off-target sites kinetically occur at later developmental stage (before the mRNA/nuclease is effectively diluted out), rendering it difficult to determine by Sanger sequencing whether or not an edit truly occurred.

RESPONSE: We have added the sentences about the limitation of off-target detection in this study (lines 136-141 in the track-changed manuscript).

2. COMMENT: Related to point #1, the authors don't provide a metric as to how they determine whether a Sanger trace reveals evidence of editing or not. There are freely available websites/software that can predict editing results (%) based on Sanger trace deconvolution (TIDE, ICE, etc.). The authors should be more clear about their judgement calls and use a more rigorous metric to quantify edited vs. un-edited samples based on empirical evidence.

RESPONSE: Thank you for this advice. As you suggested, we have re-analyzed all the obtained Sanger sequencing data by web-tools. We have performed TIDE analysis of each sequence chromatogram in all experiments, and as a result, we could obtain more robust results in Figure 1C and Supplementary Figure S4D. In Figure 2, the sequencing data from HindIII-digested products were analyzed using CRISP-ID because of its low R-squared value in TIDE analysis. We have revised the sentences in Materials and

Methods section (lines 233-241) and the related sentences in Results section (lines 60-63).

3. COMMENT: The English language and grammar of the article remains a major issue and should be appropriately revised.

RESPONSE: We are sorry for not exceeding the standards. We have revised the manuscript carefully, but we cannot notice the further points that you thought were insufficient because of the limitations of our English language ability. Therefore, we will appreciate it if you point out the sentences with crucial mistakes that affect the quality of the manuscript.

4. COMMENT: The author's response to Minor Comment #1 still remains incorrect. The modified sentence: ““the improved accuracy of HypaCas9 has expanded the range of applicable loci for the efficient generation of genetically modified animals.” continues to imply that HypaCas9 EXPANDS the range of targetable loci, which simply isn't true. HypaCas9 is still subject to the same NGG PAM requirement as wild-type SpCas9, so the ‘range of applicable loci’ remain the same (and is not expanded).

RESPONSE: We have revised the sentence as follows (lines 15-17):

“These results suggest that the improved accuracy of HypaCas9 facilitates the generation of genetically modified animals.”

REVIEWERS' COMMENTS:

Reviewer #1 (Remarks to the Author):

The authors have satisfactorily addressed all comments, with no further concerns at this point.